# Rapid Test Method for Multi-Beam Profile of Phased Array Antennas

**DOI:** 10.3390/s22010047

**Published:** 2021-12-22

**Authors:** Qingchun Luo, Yantao Zhou, Yihong Qi, Pu Ye, Francesco de Paulis, Lie Liu

**Affiliations:** 1Department of Electrical and Information Engineering, Hunan University, Changsha 410082, China; qingchun.luo@generaltest.com; 2Frontier Academic Center, Pengcheng Laboratory, Shenzhen 518102, China; yihong.qi@generaltest.com; 3General Test Systems Inc., Shenzhen 518102, China; pu.ye@generaltest.com (P.Y.); lie.liu@generaltest.com (L.L.); 4Department of Industrial and Information Engineering and Economics, University of L’Aquila, 64100 L’Aquila, Italy; francesco.depaulis@univaq.it

**Keywords:** multi-beam pattern, fast test method, phased array antenna, pattern reconstruction, antenna test method

## Abstract

The measurement of the phased array antenna (PAA) is completely different from the traditional antenna, due to its multi beam patterns. Usually, each beam pattern of the PAA needs a separate measurement, which makes the overall time extremely long. Thus, the traditional method can no longer meet the efficiency and cost requirements of new PAA measurement. In this paper, a pattern reconstruction method is proposed which significantly reduce the measurement time of multi-beam PAAs. With the known array element patterns (AEP) and theoretical weighted port excitation of the beams, any beam pattern can be predicted by measuring only a certain beam pattern, due to the element excitation coefficient (including the matching, mutual coupling, and manufacturing factors, etc.) of the specific PAA being calculated. The approach has low reconstruction error in term of beam pointing accuracy, side lobe, and co-polar and cross-polar patterns while being validated for large scanning range. Through theoretical derivation and experiments, the effectiveness of the method is verified, and the testing efficiency of the phased array antenna can be improved by 10 times or even more.

## 1. Introduction

Due to its high gain and adjustable beam pointing characteristics, phased array antennas are widely used in mobile and satellite communications, radars (for automotive sensing and unmanned aerial vehicles), as well as in military and meteorological applications [1,2,3,4,5,6].

Innovations in materials (for example, GaN, etc.) and manufacturing processes (AIP) have improved phased array antenna performance and reduced costs dramatically, allowing it to be widely used [7], which results in massive production and measurement demand of phased array antennas (PAAs). Unlike conventional antennas, large PAAs are characterized by up to thousands of beam patterns and each one needs to be measured individually. It becomes a great challenge to the traditional antenna measurement method since the measurement time of PAA will drastically increase comparing with that of conventional antennas [8,9].

Recent progress [10,11,12] has been made based on experimental environment and focused on the fast measurement of a single antenna pattern in anechoic chambers or nonanechoic chambers. Some efficient methods, known as antenna pattern interpolation, are implemented to reconstruct the 3D pattern from 2D pattern, by reducing the number of pattern cuts or sampling points being measured [13,14]. The phase-shift measurement method is also introduced in [15] to accelerate the far-field pattern calculation. The method is essentially an electrical scanning approach, substituting the traditional mechanical scanning. These works provide great convenience for antenna pattern measurement at specific angles. Nevertheless, there are only a few studies on reducing the elements or beams to be measured, which may significantly reduce the measurement time.

Pattern reconstruction or projection methods for PAAs was proposed by Herbert M. Aumann back in 1989 [16]. A theory for array calibration and far-field radiation pattern prediction using mutual coupling measurement was developed, which was complex and restricted by the self-calibration capacity of the PAA. A method which reduces the number of measured beam patterns by 35% was introduced in [17] to reconstruct the desired beam pattern, relying on a series of well-arranged beam patterns being measured (not less than the number of array elements). However, there also remains a significant error between reconstructed and actual beam patterns. Furthermore, a novel pattern reconstruction method was mentioned in [18], which allows the measured patterns not more than the array elements, thus reducing additional 20% time cost. These methods provide a good attempt to effectively reduce the testing time of multi-beam PAAs, but still face a great challenge for large PAAs, where the number of beams to be measured is still considerable.

In this paper, a model-based source reconstruction method [19,20,21,22] is proposed to predict the desired beam patterns, which greatly reduce measurement time of multi beam PPAs. The far-field pattern of a PPA is considered to be a summation of the weighted active element patterns (AEPs) [23]. Arbitrary beam pattern can be reconstructed if the actual element aperture excitation matrix (including the excitation, match, weight, manufacturing factors, etc.) and the AEPs in array (mutual coupling is considered) are all obtained. The designed AEP and the weight matrix for each beam can be easily obtained, thus the question turns to find the excitation bias between designed and actual array elements, which we call ‘excitation coefficient’, *C*. With the proposed method, only one beam pattern needs to be measured to calculate the excitation coefficient *C*, and then, arbitrary beam pattern of the PPA can be predicted accordingly. This method dramatically reduces the beam patterns measurement amount than other pattern reconstruction method, especially in the case of large array.

The rest of this paper is organized as follows. The proposed algorithm is described in Section 2. In Section 3, a 64-element PPA is used to verify the feasibility of the method, and the results and comparisons are showed here. The measurement results and comparisons are fully discussed in Section 4. Finally, conclusions and the further works are summarized in Section 5.

## 2. Theory and Methods

An array of *N* elements, as shown in Figure 1, is characterized by the far-field radiation pattern *E*(*θ*, *ϕ*) expressed in Equation (1),
(1)$$E\left(\theta ,\phi \right)=\overset{N}{\underset{i=1}{\sum }}I^{\prime }{}_{i}e^{j\varphi_{I_{i}}}{\tilde{f}}_{i}\left(\theta ,\phi \right)e^{jk\left(r_{i}\;\cdot \;{\hat{r}}_{o}\right)}$$
where ${\tilde{f}}_{i}\;\left(\theta ,\phi \right)$ is the far-field radiation pattern of the *i*-th element within the array, i.e., in the presence of all the other surrounding elements; $I^{\prime }{}_{i}e^{j\varphi_{I_{i}}^{\prime }}$ is the actual aperture excitation of the *i*-th element; $\vec{r_{o}\left(\theta ,\phi \right)}$ represents the vector from the coordinate origin *O* to the test point; $\vec{r_{i}\left(\theta ,\phi \right)}$ denotes the vector from the position of the *i*-th element to the test point. Basically, Equation (1) implies that the far-field radiation is obtained as a combination (sum) of the active element pattern ${\tilde{f}}_{i}\;\left(\theta ,\phi \right)$. The mutual coupling effects are accounted into the active element pattern ${\tilde{f}}_{i}\;\left(\theta ,\phi \right)$, whereas other non-ideal factors—such as excitation and matching of elements, manufacturing error, and T/R component differences—are considered in the aperture excitation $I^{\prime }{}_{i}e^{j\varphi_{I_{i}}}$ [24,25].

For a specific beam pattern, a theoretical weighted excitation matrix *I* at the element ports can be expressed as Equation (2), where *N* is the total number of elements. Equation (3) shows the relevance between the actual aperture excitation *I*′ and the theoretical port excitation *I*, The difference between *I*′ and *I* is defined as excitation coefficient *C*, which is determined by the non-ideal factors above, as shown in Figure 2.
(2)$$I=\left[\begin{array}{c} \begin{array}{c} I_{1}e^{j\varphi_{I_{1}}}\\ I_{2}e^{j\varphi_{I_{2}}} \end{array}\\ \vdots \\ I_{N}e^{j\varphi_{I_{N}}} \end{array}\right]$$
(3)$$I^{\prime }=\left[\begin{array}{c} \begin{array}{c} I_{1}^{\prime }e^{j\varphi_{I_{1}}^{\prime }}\\ I_{2}^{\prime }e^{j\varphi_{I_{2}}^{\prime }} \end{array}\\ \vdots \\ I_{N}^{\prime }e^{j\varphi_{I_{N}}^{\prime }} \end{array}\right]=I\;\cdot \;C=\left[\begin{array}{c} \begin{array}{c} I_{1}e^{j\varphi_{I_{1}}}\\ I_{2}e^{j\varphi_{I_{2}}} \end{array}\\ \vdots \\ I_{N}e^{j\varphi_{I_{N}}} \end{array}\right]\;\cdot \;\left[\begin{array}{c} C_{1}\\ C_{2}\\ \begin{array}{c} \vdots \\ C_{N} \end{array} \end{array}\right]$$

Definition of the AEP matrix $F_{o}$ is shown in Equation (4), and the Equation (1) can be expressed as Equation (5), combining Equations (3) and (4). The objective is to solve the excitation coefficient matrix *C* of the PAA. which can be used for arbitrary beam pattern prediction. The following describes the specific method how to obtain the excitation coefficient *C* from the far-field radiation pattern *E*(*θ*, *ϕ*) of the PAA.
(4)$$F_{o}=\left[\begin{array}{c} {\tilde{f}}_{1}\left(\theta ,\phi \right)e^{jk\left(r_{1}\;\cdot \;{\hat{r}}_{o}\right)}\\ {\tilde{f}}_{2}\left(\theta ,\phi \right)e^{jk\left(r_{2}\;\cdot \;{\hat{r}}_{o}\right)}\\ \begin{array}{c} \vdots \\ {\tilde{f}}_{N}\left(\theta ,\phi \right)e^{jk\left(r_{N}\;\cdot \;{\hat{r}}_{o}\right)} \end{array} \end{array}\right]$$
(5)$$E\left(\theta ,\phi \right)=F_{o}{}^{T}\left(I\;\cdot \;C\right)$$

Performing the far-field measurements on the PAA, with M sampling points (*M ≥ N*), then, a set of far-field measurement data is obtained, named *E**_j_* (*j* = 1, 2, …, *M*), where, *E_j_ = E*(*θ_j_*, *ϕ_j_*). Equation (6) can be easily derived from Equation (5), as below,
(6)$$\left[\begin{array}{c} \begin{array}{c} E_{1}\\ E_{2} \end{array}\\ \vdots \\ E_{M} \end{array}\right]=\left[\begin{array}{c} \begin{array}{c} {\tilde{f}}_{1}\left(\theta_{1},\phi_{1}\right)e^{jk\left(r_{11}\;\cdot \;{\hat{r}}_{o1}\right)}\\ {\tilde{f}}_{1}\left(\theta_{2},\phi_{2}\right)e^{jk\left(r_{12}\;\cdot \;{\hat{r}}_{o1}\right)} \end{array}\\ \vdots \\ {\tilde{f}}_{1}\left(\theta_{M},\phi_{M}\right)e^{jk\left(r_{1M}\;\cdot \;{\hat{r}}_{oM}\right)} \end{array}\begin{array}{c} \cdots \\ \cdots \\ \vdots \\ \cdots \end{array}\begin{array}{c} \begin{array}{c} {\tilde{f}}_{N}\left(\theta_{1},\phi_{1}\right)e^{jk\left(r_{N1}\;\cdot \;{\hat{r}}_{o1}\right)}\\ {\tilde{f}}_{N}\left(\theta_{1},\phi_{1}\right)e^{jk\left(r_{N1}\;\cdot \;{\hat{r}}_{o1}\right)} \end{array}\\ {\tilde{f}}_{N}\left(\theta_{1},\phi_{1}\right)e^{jk\left(r_{N1}\;\cdot \;{\hat{r}}_{o1}\right)}\\ {\tilde{f}}_{N}\left(\theta_{1},\phi_{1}\right)e^{jk\left(r_{N1}\;\cdot \;{\hat{r}}_{o1}\right)} \end{array}\right]\;\cdot \;\left[\begin{array}{c} \begin{array}{c} I_{1}^{\prime }e^{j\varphi_{I_{1}}^{\prime }}\\ I_{2}^{\prime }e^{j\varphi_{I_{2}}^{\prime }} \end{array}\\ \vdots \\ I_{N}^{\prime }e^{j\varphi_{I_{N}}^{\prime }} \end{array}\right]$$
where ${\hat{r}}_{oj}\;$ (*j* = 1, 2, ⋯, *M*) represents the unit vector from the coordinate origin to the *j*-th test point; $r_{ij}$ (*i* = 1, 2, ⋯, *n*; *j* = 1, 2, ⋯, *M*) represents the vector between the *i*-th element and the *j*-th test point; ${\tilde{f}}_{i}\left(\theta_{j},\phi_{j}\right)$ represents the radiation field of the *i*-th element at the *j*-th measurement point $\left(\theta_{j},\phi_{j}\right)$.

In Equation (6), the vector *E* and the matrix *F* can be defined according to Equations (7) and (8), respectively, thus the compact form in Equation (9) is obtained. The element aperture excitation $I^{\prime }$ can be calculated by multiplying the term ${\left(F^{T}F\right)}^{-1}F^{T}$ at both side of Equation (9), as shown in Equation (10).
(7)$$E=\left[\begin{array}{c} E_{1}\\ E_{2}\\ \begin{array}{c} \vdots \\ E_{M} \end{array} \end{array}\right]$$
(8)$$F=\left[\begin{array}{c} \begin{array}{c} {\tilde{f}}_{1}\left(\theta_{1},\phi_{1}\right)e^{jk\left(r_{11}\;\cdot \;{\hat{r}}_{o1}\right)}\\ {\tilde{f}}_{1}\left(\theta_{2},\phi_{2}\right)e^{jk\left(r_{12}\;\cdot \;{\hat{r}}_{o2}\right)} \end{array}\\ \vdots \\ {\tilde{f}}_{1}\left(\theta_{M},\phi_{M}\right)e^{jk\left(r_{1M}\;\cdot \;{\hat{r}}_{oM}\right)} \end{array}\begin{array}{c} \begin{array}{c} \ldots \\ \ldots \end{array}\\ \ddots \\ \ldots \end{array}\begin{array}{c} \begin{array}{c} {\tilde{f}}_{N}\left(\theta_{1},\phi_{1}\right)e^{jk\left(r_{N1}\;\cdot \;{\hat{r}}_{o1}\right)}\\ {\tilde{f}}_{N}\left(\theta_{2},\phi_{2}\right)e^{jk\left(r_{N2}\;\cdot \;{\hat{r}}_{o2}\right)} \end{array}\\ \vdots \\ {\tilde{f}}_{N}\left(\theta_{M},\phi_{M}\right)e^{jk\left(r_{NM}\;\cdot \;{\hat{r}}_{oM}\right)} \end{array}\right]$$
(9)$$E=FI^{\prime }=F\left(I\;\cdot \;C\right)$$
(10)$${\left(F^{T}F\right)}^{-1}F^{T}E={\left(F^{T}F\right)}^{-1}F^{T}FI^{\prime }=I^{\prime }=\left(I\;\cdot \;C\right)$$

So far, the *M* far-field data *E* can be obtained with a normal 3D radiation pattern measurement, and the radiation pattern *F* of the array elements can be obtained by simulation or actual measurement, thus the aperture field excitation $I^{\prime }$ of the element can be easily calculated. When *M* = *N*, the number of equations is equal to the number of unknowns to be solved, the aperture field excitation $I^{\prime }$ can be obtained by simply solving a linear equation system. In the case of *M > N*, the number of equations is greater than the number of unknowns, the aperture field excitation $I^{\prime }$ can be obtained by applying the least square method. Since the ideal weighted port excitation *I* for a specific beam pattern is assumed to be known (theoretical value), the excitation coefficient matrix *C* can be readily obtained. With the coefficient *C*, the radiation pattern $E_{x}$ for any beam pattern can be obtained by applying Equation (11).
(11)$$E_{x}=FI_{x}^{\prime }=F\left(I_{x}\;\cdot \;C\right)$$

Following this procedure, all the beam patterns of the PAA can be readily predicted, relying only on a normal radiation pattern measurement.

In order to verify the feasibility of the proposed method, a 64-element PAA is tested inside the far-field anechoic chamber.

The validation steps are as follows:
Install the PAA on the test system turntable and adjust the center of the antenna to coincide with the center of the rotation system,Use the beam controller to adjust the PAA to a certain main beam orientation ($\theta_{0},\;\varphi_{0}$) and test the radiation pattern $E_{0}$,Obtain the weighted element port excitation *I* of the designated beam orientation ($\theta_{0},\;\varphi_{0}$),Obtain the simulation radiation pattern *F* of all the elements in the array with the coupling factor being considered,Use Equation (10) to calculate the excitation coefficient *C* of this antenna,Obtain the element port excitation $I_{x}$ at the beam orientation ($\theta_{x},\varphi_{x}$), which is desired to be predicted,Calculate the radiation pattern $E_{x}$ of the beam orientation ($\theta_{x},\;\varphi_{x}$) using Equation (11),Adjust the PAA to the beam orientation ($\theta_{x},\varphi_{x}$) with the wave controller, and measure the actual radiation pattern $\tilde{E_{x}}$ of this beam orientation,Compare $E_{x}$ and $\tilde{E_{x}}$.

## 3. Results

A 64-element PAA was measured to validate the proposed fast test method in a spherical range chamber with a measurement distance of 2.5 m, as shown in Figure 3. The PAA size is about 80 × 80 mm and the test frequency is 25 GHz, thus the distance meets the far-field criteria $\frac{2D^{2}}{\lambda }$.

The measured beam orientation at Step 2 is designated as ($\theta_{0},\;\varphi_{0}$) = (0, 0), with a sampling range of *φ* from 0 to 180°, θ from −90° to 90°, and a sampling step of 2°. Based on the simulated radiation pattern *F* obtained at Step 4 and the ideal port excitation *I* of all the elements in the array, the excitation coefficient *C* of the elements is obtained by using Equation (10), according to Step 5. Calculated the radiation patterns of arbitrary three beam orientation, such as (*θ*, *φ*) at (15, 0), (30, 0), (60, 0), by Equation (11). The comparison of predicted and measured radiation pattern are shown in Figure 4, Figure 5 and Figure 6. A detailed analysis of the main polarization and cross polarization pattern bias is carried out and reported in Table 1, Table 2 and Table 3 for the three designated cases.

## 4. Discussion

The comparison of the data above in Figure 4, Figure 5 and Figure 6 and in Table 1, Table 2 and Table 3 show that the predicted and measured radiation patterns agree well within the range of both the primary and secondary lobes at different beam orientation (*θ*, *φ*), even for large scanning angles, i.e., (*θ*, *φ*) = (60, 0). At boresight, the bias of beamwidth is less than 2.5% and the predicted beam orientation agrees extremely well with the measured and also the theoretical beam orientation, with a bias of about 0.01° to 0.03°; For the side lobe which is about 15 dB lower than the boresight, the max deviation between the predicted and measured data is no more than 0.75 dB; This method also provides a excellent match at a cross-polarization ratio of about −25 dB, where the bias is and less than 2.71 dB. Considering the measurement uncertainty of the PAA, the pattern prediction accuracy is well enough to replace the one-by-one measurement [26,27,28,29].

For example, a full beam patterns measurement for the 64-element PAA covers a range of *θ* = 0°~60° and *φ* = 0°~360°, with 10° angular interval. The total number of the beam patterns to be measured is 217. The measurement time for a single beam pattern is about 30 min. It will take 6510 min to complete all the beam pattern measurements one by one with the traditional method. With the pattern reconstruction method mentioned in [17,18], the total measurement time may be reduced to 40%—i.e., 2604 min—in a good case. With the source reconstruction method in this paper, it takes about 1 min for coefficient C calculating, 30 min for a certain beam pattern measurement and 0.5 min for each beam pattern projection, resulting as a total time cost of 139 min. A comparison of the efficiency of the fast test method and the conventional method is shown in Table 4.

The comparison above shows that the proposed source reconstruction method can greatly improve the measurement efficiency of PAAs. Additionally, it provides an excellent match at the boresight and the first side lobes, both co-polar and cross-polar, which are mainly considered in general. On the other hand, a larger deviation is found at the far side lobes, which is presumed to be caused by the combined effect of the following factors: the difference between the simulation and the actual radiation pattern of the array elements, the installation position deviation of the antenna, the uncertainties of the test system and of the nominal excitation values *Ix* of each element. These issues lead the aperture field excitation coefficient matrix *C* deviating from the realistic value.

## 5. Conclusions

In this paper, a multi-beam PAA fast test method based on a source reconstruction algorithm is proposed. Through theoretical derivation and experimental verification, the aperture field excitation coefficient, *C*, of a PAA can be calculated through any beam pattern measurement, and then an arbitrary beam pattern can be constructed with the proposed method. The reconstructed beam pattern agrees very well with the measured one for the main lobe, although slightly greater error was found from the side lobes. This method avoids testing the beam patterns individually, thus greatly improving the testing efficiency of PAAs.

For a PAA with dozens to even thousands of beam patterns to be measured, the conventional method can only measure each beam pattern individually. However, by using the method proposed in this paper, only one beam pattern measurement is needed to obtain the total desired beam patterns with acceptable accuracy, which greatly reduces the testing time of a multibeam PAA. Especially, the proposed method becomes obviously more effective for both large PAAs and the mass production measurement of the PAAs.

A further improvement of the proposed method will be carried out in the future by analyzing the accuracy when more than one measured beam pattern is used to find more accurate solutions for the excitation coefficient matrix *C*. Further research will also be conducted for the scenarios when the element radiation patterns in the array are not known.

## Figures and Tables

**Figure 1 sensors-22-00047-f001:**
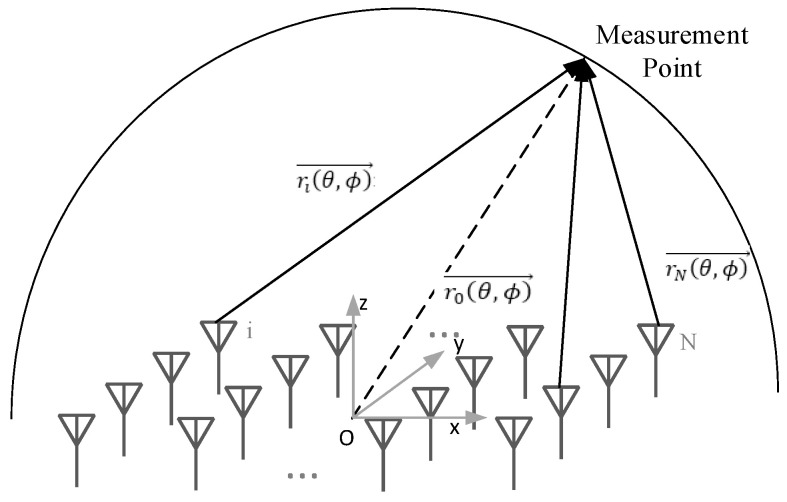
Schematic diagram of a *N*-element phased array.

**Figure 2 sensors-22-00047-f002:**
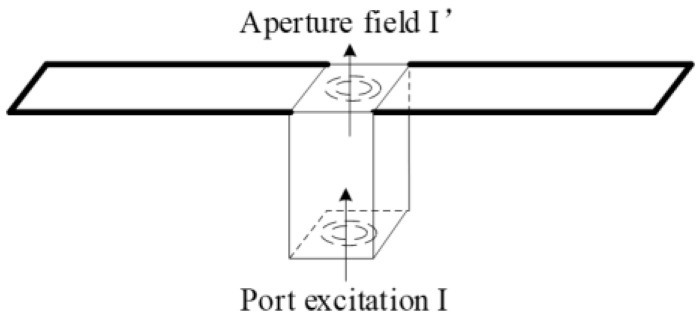
Schematic diagram of port excitation and aperture excitation of an element.

**Figure 3 sensors-22-00047-f003:**
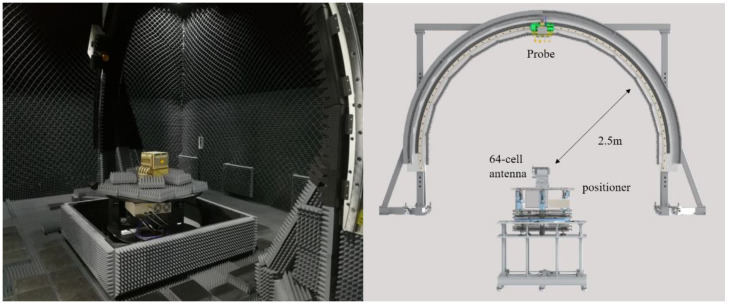
The 64-element PAA measured in a spherical range measurement system.

**Figure 4 sensors-22-00047-f004:**
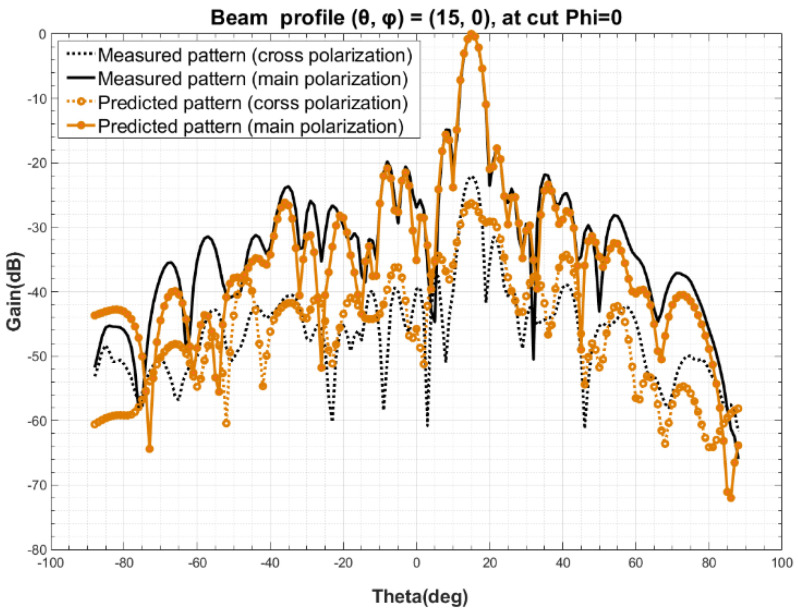
Comparison of predicted and measured beam profile at (*θ*, *φ*) = (15, 0).

**Figure 5 sensors-22-00047-f005:**
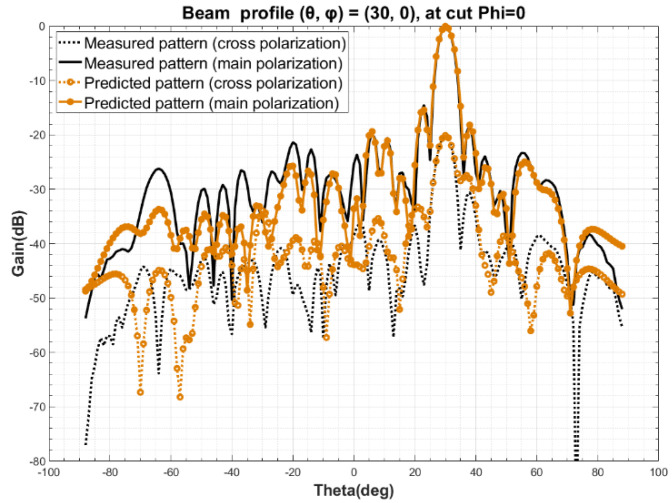
Comparison of predicted and measured beam profile at (*θ*, *φ*) = (30, 0).

**Figure 6 sensors-22-00047-f006:**
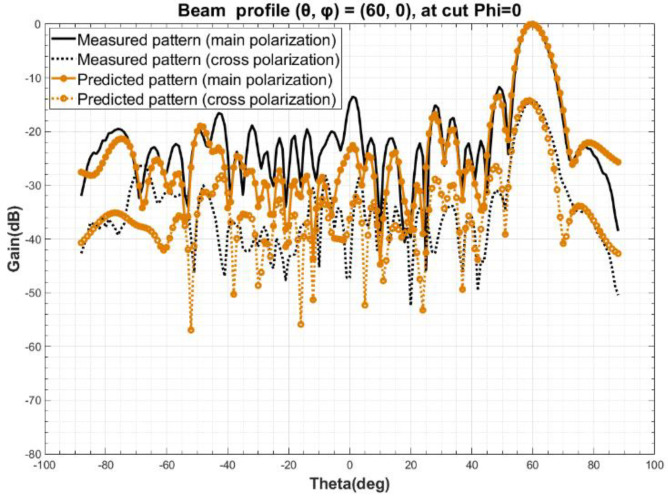
Comparison of predicted and measured beam profile at (*θ*, *φ*) = (60, 0).

**Table 1 sensors-22-00047-t001:** Detailed factor comparison of predicted and measured beam profile at (*θ*, *φ*) = (15, 0).

(15, 0)	3 dB Beam Width (°)	Beam Orientation Accuracy *θ* (°)	First Left Side Lobe Peak (dB)	First Right Side Lobe Peak (dB)	Cross-Pol. (dB) ^1^
Measured	4.18	15.01	−14.58	−18.19	19.57
Predicted	4.25	15.02	−15.33	−18.17	19.69
Deviation	0.07	0.01	−0.75	0.02	0.12

^1^ The cross-polarization is the maximum difference between the main polarization and the cross-polarization within a 3 dB beamwidth.

**Table 2 sensors-22-00047-t002:** A detailed factor comparison of predicted and measured beam profile at (*θ*, *φ*) = (30, 0).

(30, 0)	3 dB Beam Width (°)	Beam Orientation Accuracy *θ* (°)	First Left Side Lobe Peak (dB)	First Right Side Lobe Peak (dB)	Cross-Pol. (dB) ^1^
Measured	4.54	30.01	−14.94	−17.44	−21.75
Predicted	4.66	30.01	−15.55	−17.76	−24.46
Deviation	0.12	0	0.61	0.32	2.71

^1^ The cross-polarization is the maximum difference between the main polarization and the cross-polarization within a 3 dB beamwidth.

**Table 3 sensors-22-00047-t003:** Detailed factor comparison of predicted and measured beam profile at (*θ*, *φ*) = (60, 0).

(60, 0)	3 dB Beam Width (°)	Beam Orientation Accuracy *θ* (°)	First Left Side Lobe Peak (dB)	First Right Side Lobe Peak (dB)	Cross-Pol. (dB) ^1^
Measured	7.84	59.99	−11.79	−22.62	−14.01
Predicted	8.11	59.97	−13.21	−22.07	−12.95
Deviation	0.27	0.02	1.42	0.55	−1.06

^1^ The cross-polarization is the maximum difference between the main polarization and the cross-polarization within a 3 dB beamwidth.

**Table 4 sensors-22-00047-t004:** Efficiency comparison of different method.

Item	Source Reconstruction Method	Pattern Reconstruction Method	Conventional Method
Coefficient C calculating (min)	1	0	0
Beam patterns measured number	1	80 (EST)	217
Beam patterns measured time (min)	30	2400	6510
Beam patterns projection time (min)	108	204 (EST)	-
Total duration (min)	139	2604	6510
Time cost reduction (%)	97%	40%	-

## Data Availability

Restrictions apply to the availability of these data. Data was obtained from CASC 504 Institute and are available from the authors with the permission of CASC 504 Institute.

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
