# Peer review of "Rapid Test Method for Multi-Beam Profile of Phased Array Antennas"

_sensors, 2021, doi:10.3390/s22010047_

Round 1
Reviewer 1 Report
The authors presented a test method for the 5G phased array antennas. The idea and work are quite interesting. The reviewer has the following comments.
1) The Introduction (literature review) of the manuscript is poor. The current references are not enough to locate this work in state-of-the-art. There are many related papers (even the papers on this topic have been published in the submitted special issue). Authors are advised to improve the Introduction section by a rigorous literature review.
2) The presentation and the arrangement of the data are poor. The abstract and the conclusion need improvements. For example, there is no data on how much test time has been reduced in this work, and how much accuracy has been compromised.
3) The reviewer would like to see a performance comparison table (in terms of test time, efficiency, test type) with other OTA testing methods.
4) The manuscript should be revised for grammatical errors and better readability.
Author Response
REVIEWERS' COMMENTS:
Reviewer: 1
Comments for Transmittal to Author
The authors presented a test method for the 5G phased array antennas. The idea and work are quite interesting. The reviewer has the following comments:
Point 1: The Introduction (literature review) of the manuscript is poor. The current references are not enough to locate this work in state-of-the-art. There are many related papers (even the papers on this topic have been published in the submitted special issue). Authors are advised to improve the Introduction section by a rigorous literature review.
Response 1: Thanks for your comment. We have extensively revised the introduction and supplemented it with some literature on related topics, outlining the methods and the principles and effects of the methods described in these articles for improving the efficiency of antenna testing. In the introduction, the principle of the method proposed in this paper is also described in detail, as well as the differences with other fast testing methods, and the advantages of the proposed method and the results achieved are presented.
Point 2: The presentation and the arrangement of the data are poor. The abstract and the conclusion need improvements. For example, there is no data on how much test time has been reduced in this work, and how much accuracy has been compromised.
Response 2: We have added and improved the abstract and conclusions, which detailed the principles and measurement efficiency comparisons between the proposed method and other fast test methods. We also improved and added some graphical content, in addition to a detailed description of the test accuracy and applicability of this method in the discussion.
Point 3: The reviewer would like to see a performance comparison table (in terms of test time, efficiency, test type) with other OTA testing methods.
Response 3:We have added descriptions and tables comparing the performance between the methods in this paper and other methods to the Section 4.
Point 4: The manuscript should be revised for grammatical errors and better readability.
Response 4: We have made extensive presentation and grammatical changes to the full text, details of which can be found in the full revised manuscript.
Reviewer 2 Report
In this paper, the efficiency of traditional detection of active phased array antenna radiation performance is presented. By using the array simulated element pattern and the ideal port excitation, the beam profile in any direction can be predicted by testing a certain beam profile, and the method is verified by theoretical derivation and experiment. The paper has clear logic and structure. But there are still some problems that need to be improved.
1.In the abstract, you should clearly state what your methods and techniques are.
2.The English expression of the whole paper needs to be strengthened.
3.In the paragraph, there are problems with the description of the equation. For example,”... expressed in (1)”on line 66 is inappropriate. The correct expression is “Expressed in eq. (1)”.
4. Pay attention to the layout of the table.
5. The papershould include some comparisons of results and analyses with existing methods.
6. Both the introduction and the experimental analysis are inadequate and should be expanded.
Author Response
REVIEWERS' COMMENTS:
Reviewer: 2
Comments for Transmittal to Author
In this paper, the efficiency of traditional detection of active phased array antenna radiation performance is presented. By using the array simulated element pattern and the ideal port excitation, the beam profile in any direction can be predicted by testing a certain beam profile, and the method is verified by theoretical derivation and experiment. The paper has clear logic and structure. But there are still some problems that need to be improved.
Point 1: In the abstract, you should clearly state what your methods and techniques are.
Response 1: Thanks for your comment. We have added and improved the abstract. The proposed methods and techniques are discribed in details.
Point 2: The English expression of the whole paper needs to be strengthened.
Response 2: We have made extensive presentation and grammatical changes to the full text, details of which can be found in the full revised manuscript.
Point 3: In the paragraph, there are problems with the description of the equation. For example,”... expressed in (1)”on line 66 is inappropriate. The correct expression is “Expressed in eq. (1)”.
Response 3: Thanks for your suggestion. The description of the equations are all changed to Eq. (X).
Point 4: Pay attention to the layout of the table.
Response 4: Thanks for your comment. We have revised the layout of the table to meet the formatting requirements of the journal.
Point 5: The paper should include some comparisons of results and analyses with existing methods.
Response 5: We have added the analyses with the existing methods in the Section 2, Introduction. The comparisons of proposed method and other fast test method are added in the Section 4, Discussion.
Point 6: Both the introduction and the experimental analysis are inadequate and should be expanded.
Response 6: Thanks for your comment. We have rewrite the introduction in the Section 2, added the analysis of other fast test mothods, and expanded the discussion in the Section 4 with comparision of the test methods.
Round 2
Reviewer 1 Report
The authors have revised the manuscript well. All comments of the Reviewer have been addressed.